# Extraction Techniques and Analytical Methods for Isolation and Characterization of Lignans

**DOI:** 10.3390/plants11172323

**Published:** 2022-09-05

**Authors:** Andrzej Patyra, Małgorzata Kołtun-Jasion, Oktawia Jakubiak, Anna Karolina Kiss

**Affiliations:** 1Department of Pharmacognosy and Molecular Basis of Phytotherapy, Medical University of Warsaw, 02-097 Warsaw, Poland; 2Doctoral School, Medical University of Warsaw, 02-091 Warsaw, Poland; 3Institut des Biomolécules Max Mousseron, Université de Montpellier, CNRS, ENSCM, 34293 Montpellier, France

**Keywords:** lignans, extraction, HPLC, MS, TLC

## Abstract

Lignans are a group of natural polyphenols present in medicinal plants and in plants which are a part of the human diet for which more and more pharmacological activities, such as antimicrobial, anti-inflammatory, hypoglycemic, and cytoprotective, are being reported. However, it is their cytotoxic activities that are best understood and which have shed light on this group. Two anticancer drugs, etoposide, and teniposide, were derived from a potent cytotoxic agent—podophyllotoxin from the roots of *Podophyllum peltatum*. The evidence from clinical and observational studies suggests that human microbiota metabolites (enterolactone, enterodiol) of dietary lignans (secoisolariciresinol, pinoresinol, lariciresinol, matairesinol, syringaresinol, medioresinol, and sesamin) are associated with a reduced risk of some hormone-dependent cancers. The biological in vitro, pharmacological in vivo investigations, and clinical studies demand significant amounts of pure compounds, as well as the use of well-defined and standardized extracts. That is why proper extract preparation, optimization of lignan extraction, and identification are crucial steps in the development of lignan use in medicine. This review focuses on lignan extraction, purification, fractionation, separation, and isolation methods, as well as on chromatographic, spectrometric, and spectroscopic techniques for their qualitative and quantitative analysis.

## 1. Introduction

Lignans are a group of natural polyphenols (and one of the most lipophilic), located chiefly in the plant cell walls, many of which are present in medical plants and in plants that are a part of the human diet. According to the recent nomenclature, lignans are dimers of two coniferyl, sinapyl, paracoumaryl, or similar alcohol monomers. Some authors restrict the term lignan only to those molecules coupled by the central carbon of the side-chain (i.e., 8,8′ or β,β’ dimers). In this manner, there is a distinction between neolignans—similar compounds coupled differently (by another carbon or through oxygen bonds), norlignans—lignans which lost one of their carbon atoms from their skeleton, lignoids—compounds that have undergone skeletal rearrangement, and oligolignans—lignan condensation products—which we can divide into sequilignans (trimers) and dilignans (tetramers) [1]. Lignans were found in all of the plant parts, though their highest content was reported in plant wood—especially in wood knots—roots, bark, leaves, flowers, fruits, and seeds [2]. Lignans’ and neolignans’ formation are subject to stereoselective biosynthesis (Figure 1); thus, they are enantiomerically pure [1]. They mostly appear linked with carbohydrates, while more lipophilic aglycon-free forms can be found in wood and bark [3]. 

There is a rising interest in lignans and plants rich in this class of polyphenols as more and more pharmacological activities are being reported, such as antimicrobial, anti-inflammatory, hypoglycemic, and cytoprotective. However, it is their cytotoxic activities that are best understood, and which have shed light on this group. Two anticancer drugs, etoposide and teniposide, were derived from a potent cytotoxic agent—podophyllotoxin from the roots of *Podophyllum peltatum* L. This led to the search for other cytotoxic metabolites among these polyphenols, with interesting finds concerning lignans and neolignans from nearly every structural subtype [12]. Interestingly, the evidence from clinical and observational studies suggests that the human microbiota metabolites (enterolactone, enterodiol) of dietary lignans (secoisolariciresinol, pinoresinol, lariciresinol, matairesinol, syringaresinol, medioresinol, and sesamin) are associated with reduced risk of some hormone-dependent cancers [13]. Primarily, the attention is focused on flaxseeds’ lignan—secoisolariciresinol diglucoside (SDG) and the effect on breast and colorectal cancer development [14,15].

Moreover, some in vitro and in vivo studies suggest that arctiin and its aglycon arctigenin, occurring in the Great Burdock fruits and *Forsythia* spp., are promising cytotoxic agents. Arctigenin was shown to induce tumor cell death, such as prostate, breast, lungs, liver, and colon, by inducing apoptotic signaling pathways, caspases, cell cycle arrest, and the modulating proteasome [16]. Other reports suggest the potential antiproliferative effect of lignans from *Phyllanthus* spp., *Schisandra chinensis* (Turcz.) Baill. or *Magnolia officinalis* Rehder & E.H. Wilson [17,18,19].

The biological in vitro, pharmacological in vivo investigations, and clinical studies demand significant amounts of pure compounds, as well as the use of well-defined and standardized extracts. That is why proper extract preparation, optimization of lignan extraction, and identification are crucial steps in the development of lignan use in medicine.

This review focuses on the lignan extraction, purification, fractionation, separation, and isolation methods, as well as on chromatographic, spectrometric, and spectroscopic techniques for their analysis. Considering that lignans and similar compounds are abundant in plants, and each year many new compounds are isolated, we decided to devote this paper to plant materials where the lignans are present in more significant amounts and adopt the strict definition of lignans mentioned before. This does not mean that the methods and techniques presented here do not apply to neolignans, norlignans, lignoids, or oligolignans’ isolation and analysis, as they may be present in the same plant material as true lignans. Still, our attention will not be directed to their physiochemical properties, and further considerations should be taken in relation to them.

A literature search for the present review was conducted using the following databases: PubMed, Scopus, and Google Scholar, with the following search terms: “lignans”, “extraction”, “isolation”, “detection”, “quantification”, and “analysis” in various combinations. The results were first screened for their relevance based on their abstract and afterwards the full texts were analyzed. Studies from the year 1998–2022 were considered, as this period was characterized by considerable interest in lignan analysis. The search was conducted on 4 July 2022.

## 2. Sample Preparation

The isolation of plant metabolites begins with the collection and handling of plant material. In this regard, the procedure is typical for isolating any plant metabolite and should include botanical or taxonomical identification and preparation and storage of voucher specimens [20]. The plant material that is otherwise acquired should have its origin noted, and certificates should be provided if possible. When collecting plant material, it should be considered in which plant parts the target compounds are accumulated. For instance, in wood, the lignans are primarily present in knot wood (up to 30% (w/w) of extract) than in heartwood (less than 0.1%), and so the separation of knot wood from other wood parts may prove beneficial [21,22]. Similarly, lignans are found in the hulls of seeds rather than in their embryos. Thus, isolation can be facilitated by hulling the seeds, i.e., separating the hulls from kernels [23].

Every plant material should be handled and stored in a way that will provide stability to plant metabolites; thus, they will be protected from transformation and degradation. In most cases, the material will be air-dried, oven-dried, or freeze-dried [24]. Lignans and their glycosides are relatively resistant to high temperatures, so the options are numerous. Both freeze-drying methods [25] and drying in temperatures as high as 60 °C [26] were successfully applied. 

Higher temperatures have been applied to study the temperature stability of lignans in plant material and food products. Generally, lignans and their aglycones from flaxseed, sesame seed, and rye seem stable below the temperature of 100 °C, while applied heat also facilitated the extraction procedure [27]. The stir-frying of Great Burdock fruit at 150 °C led to changes in the lignan content, predominantly manifesting a higher aglycones’ and lower glycosides’ yield, which could be explained by the enhancement of hydrolysis processes [28,29]. Some aglycones (e.g., pinoresinol, sesamin) are stable in temperatures as high as 180–200 °C [30]. Similarly, the secoisolariciresinol diglucoside ester-linked complex with hydroxymethyl glutaryl (found in flaxseed) is stable even in bakery products [27]. The overall thermal stability of lignans depends on the compound structure and its interactions with the other compounds present in the plant matrix [27,30]. Although heating may positively impact on the extractability of lignans, the loss of some less temperature-stable compounds (i.e., neolignans linked by oxygen or sesquilignans), and the possible hydrolysis of glycosides, should always be brought into consideration.

Not many studies have focused on the photostability of lignans. For instance, Kawamura et al. [31] light-irradiated 7-hydroxymatairesinol in different media (chloroform, methanol, water, and directly in plant matrix) and identified oxidation products, such as α-conidendrin, 7-oxomatairesinol, allohydroxymatairesinol, 7-methoxymatairesinol, and vanillin. Despite the scarcity of data on the photostability of lignans, the protection of the plant material and isolated compounds from light is reasonable. 

Lastly, sometimes lignans can be stored in plants in a macromolecular complex, such as secoisolariciresinol diglucoside in flaxseed. It is esterified with 3-hydroxymethylglutaric acid and other phenolic compounds, such as p-coumaric acid and ferulic acid glycosides [32]. Various methods have been implemented to facilitate the extraction, such as acidic, alkaline, and enzymatic hydrolysis [33]. Although acidic hydrolysis can efficiently break the ester and glycoside linkages, liberating secoisolariciresinol, the product is not stable in these conditions and can transform to anhydrosecoisolariciresinol [34].

## 3. Extraction

The extraction process is a crucial and often overlooked step in isolating secondary metabolites. Several parameters influencing the extraction yield should be considered for the optimum recovery of lignans from the plant matrix. These include the choice of extraction solvent or solvents, method, time, temperature, solvent-to-sample ratio, and the number of repeat extractions of a sample [24]. 

## 4. Solvents

The choice of extraction solvent should be based on its physicochemical properties (polarity, solubility, lipophilicity), safety, ease of use, the potential for artifact formation, grade and purity, selectivity, and cost. The solvent should be matched with the target compound for the physicochemical properties. In this case, lignans are fairly lipophilic polyphenols (logP = 3.3 ± 1.0) with a limited water solubility (logS = −4.2 ± 1.4), whilst lignans glycosides tend to be more hydrophilic (logP = 0.4 ± 1.1; logS = −2.5 ± 1.3) [35]. Thus, for plant material containing aglycones, medium polarity solvents, such as ethyl acetate, may be used, as well as polar solvents such as ethanol, methanol, and their aqueous mixtures. Although some less polar lignans may be extracted with less polar solvents, e.g., dichloromethane, chloroform, or even *n*-hexane, they seem to be rarely used in the primary extraction from the plant matrix and are rather found in further steps, i.e., partitioning, separation, and isolation [26,36].

Most of the lignan glycosides will not be extracted by medium polarity solvents, and the ones that are more polar may be hard to obtain using pure ethanol or methanol. In this case, aqueous mixtures of ethanol and methanol give the best results. In the case of very polar lignan glycosides, pure water may be used [37,38,39,40].

It is generally advised to select a solvent by assessing the lipophilicity of the target lignan. When planning to isolate an unknown compound or obtain more than one compound (e.g., both aglycones and glycosides), choosing aqueous mixtures of ethanol or methanol is advisable. Aqueous alcohols have the advantage over pure solvents of greatly facilitating the penetration of the solvents into the plant matrix with even a small addition of water (5–10%). The mixtures of water and alcohols have an advantage over sequential extraction with those solvents separately of lowering the water surface tension, reducing the polarity of water, and increasing its density [41].

Most commonly, concentrations of 70–100% of either aqueous ethanol or methanol are used for lignans and lignans glycosides extraction (Table 1). As shown in the table, the concentration of ethanol or methanol used does not depend on the lignan form in the plant matrix. We observed both pure alcohols’ use in glycosides’ extraction [42,43,44,45] and water’s use in free aglycones’ extraction [39].

The choice between methanol and ethanol seems to depend mainly on the practice of a particular laboratory. It is of course worth noting that methanol has certain advantages, such as higher polarity, lower boiling point (which facilitates evaporation of solvent), and lower water impurity compared to ethanol. On the other hand, it is much more toxic than ethanol, and thus ethanol is more commonly used for the isolation of compounds for biological studies and when traditionally prepared extracts are studied [92].

The presence of other substances in the plant matrix is not without influence on the extraction of lignans. In particular, some lipophilic components, such as resins, terpenoids, and fatty acids, may cause some difficulties. Thus, it is advised to use sequential extraction with a non-polar solvent for lignan sources rich in lipophilic contents, e.g., flaxseeds, sesame seeds, Burdock fruit, or conifer wood. Petroleum ether and *n*-hexane are the most commonly used solvents for defatting extracts, although pentane and dichloromethane may also be used for this purpose. Caution is advised when using dichloromethane or similar medium polar solvents, as some more lipophilic lignans, e.g., the lignans from *Magnolia* spp. L. flowers, can be extracted in those solvents [93]. Other methods may also be applied, such as lipid precipitation in 10% acetone in −40 °C [26]. The removal of fatty contents can be executed directly on the plant material or crude extract. 

Nowadays, an essential factor during solvent and method consideration is its environmental impact; thus, much attention is paid to reducing organic solvent use. Recently green extraction methods had been applied in lignan isolation from *Eleutherococcus senticosus* (Rupr. & Maxim.) Maxim. root. The authors reported using choline chloride and a lactic acid mixture in a 1:2 ratio with a 20% addition of water, successfully extracting lignan glycosides from plant matrix [94]. Although it is the first and only use of deep eutectic solvent in lignan extraction, the development of this method in the future should be expected.

In laboratory practice, solid to liquid ratios of 1:5 to 1:20 are generally considered appropriate. Higher ratios are not cost-efficient and should be used only to extract very valuable compounds. Thus, microscale experiments, which can sometimes offer ratios as high as 1:200, are not applicable on a larger scale [95]. Table 1 shows the solvent-to-sample ratios used in lignans extraction. 

## 5. Methods

Lignans’ extraction can be executed using a variety of techniques. Commonly, the conventional procedures such as Soxhlet, maceration, and digestion, were applied. However, the use of other methods, such as ultrasound-assisted extraction (UAE), accelerated solvent extraction (ASE), microwave-assisted extraction (MAE), and supercritical fluid extraction (SFE) is increasing. The use of these methods is reviewed in Table 1.

The conventional methods are often techniques of choice because they do not require complex apparatus and because of their simplicity while producing satisfactory extraction rates. On the other hand, these methods carry essential disadvantages, such as long extraction times and the use of large volumes of solvents [96]. Contrary to other conventional methods, maceration is performed at room temperature, which is why it is mainly used to extract thermolabile compounds. Although macerations as short as 1 h have been performed in lignans’ extraction [70], they usually require much longer extraction times, reaching up to 7 days [60], with 24 h commonly applied [57,59,72]. Moreover, the extraction of phenolics (and by that lignans) by maceration is relatively weak [79]. As we have discussed before, lignans are not thermolabile compounds, and thus heat may be applied to shorten the extraction procedure. For instance, Kwon et al. [83] applied gentle heat of 50 °C in the maceration of *Schisandra chinensis* (Turcz.) Baill. fruit (a technique called digestion). Similar digestion procedures were used by Sicilia et al. [34] and Frishe et al. [72] in flaxseed lignans’ extraction. 

Soxhlet and other heated reflux techniques are the most common methods used in lignans’ extraction. In their papers, the authors did not include the information on the temperature they used in their methods. However, as Zhang et al. pointed out, we can assume that it usually is performed at 80–100 °C—a temperature capable of boiling ethanol, methanol, and their dilutions, as well as other organic solvents and water [96]. The main advantage of this method is limiting the use of large volumes of solvents, as one batch of solvent is constantly recycled [92]. The time of extraction under this method varies from the 30 min used by Yang et al. in the extraction of eleutheroside E (syringaresinol diglucoside) from *Eleutherococcus senticosus* (Rupr. & Maxim.) Maxim. root [37], up to 10 h which was applied by Kumar et al. in the extraction of sesamin and sesamolin from the sesame seeds [26]. 

Sequential extraction at elevated temperature and pressure in short time periods, called accelerated solvent extraction (ASE), was applied several times in lignans’ extraction from wood knots. Willfor et al. were probably the first ones to utilize this technique in the extraction of lignans from *Picea abies* (L.) H.Karst. [97] and *Pinus sylvestris* L. wood knots [80]. The use of ASE followed studies on other conifer species, such as *Picea* spp. Mill. [98], *Abies* spp. Mill. [99], *Larix* spp. Mill., *Thuja* spp. L., *Tsuga* spp. Carrière, and *Pseudotsuga* spp. Carrière [100]. It was not seen outside this application.

Alternative techniques were implemented to overcome the disadvantages of conventional methods, such as solvent and time consumption, and environmental impact. Their common characteristic is to increase the efficiency of extraction using auxiliary energies (microwaves, ultrasounds) in place of conventional heating, stirring, shaking, or vortexing [101]. These methods are more complex than conventional ones, as the extraction yield is not only the result of solvent choice, time, temperature, solvent-to-sample ratio, and the number of repeat extractions of a sample; it is also reliant on specific parameters of sonification (wave frequency and distribution, probe or bath system) or microwave techniques (microwave power) [24]. While the advantages of these techniques are numerous, there is a risk that ultrasounds and microwaves may alter lignans’ and other metabolites’ structures [101]. 

Ultrasound-assisted extraction (UAE) was used in the recovery of lignans from *Linum usitatissimum* L. seeds, *Sesamum indicum* L. seeds, *Arctium lappa* L. fruit, and *Forsythia* spp. Vahl fruits, stems, and roots [29,30,42,62,67,68,74,75,76,88]. The extraction times were much shorter than using conventional methods, ranging from 5 min for *Sesamum indicum* L. seeds [30], to 90 min for *Forsythia viridissima* Lindl. roots [68]. However, for other parameters of sonification, only the authors of 2 papers included information on wave frequency used in their experiments—40 kHz [88] and 70 kHz [67]. Similarly, in most works, little was said about the sonification system (bath or probe) and the apparatus used. Perhaps an important note should be made on the temperature effect in ultrasound-assisted extraction. Contrary to conventional methods, a higher temperature may lower the extraction yield, as ultrasound waves will not distribute properly in a boiling solvent. For instance, Mekky et al. [88] performed their sonification experiment for sesame seeds at room temperature, while Corbin et al. [75] studied the temperature range of 25 to 60 °C during the sonification of flaxseeds and showed the lowest temperature to be superior. UAE sequential extraction method developed by Mekky et al. for lignan extraction from sesame seed cake and flaxseed cake afforded the obtention of lignans without the hydrolysis of glycosides, thus up to four hexosides’ conjugates (sesaminol tetrahexosides) were observed in the sesame seed cake extract and up to three hexosides conjugates (secoisolariciresinol trihexoside dihydroxyethylglutaryl esters) in the flaxseed cake extract [76,88]. 

Another important method is microwave-assisted extraction (MAE), which was used for the first time in lignan extraction by *Beejmohun* et al. in their study of flaxseeds [71]. A valuable work on the optimization of this method’s conditions for the preparation of lignan-rich extract from *Saraca asoca* (Roxb.) Willd. bark was completed by Mishra and Aeri [82]. According to the authors, their work concludes that microwave-assisted extraction is superior to conventional methods for the isolation of lignans—in their case, it was lyoniside, a lignan glycoside. Regarding these conditions, 70% methanol dilution was chosen as the best extraction solvent, 10 min as the extraction time, and 1:30 as the sample-to-solvent ratio. A similar optimization was completed by Lu et al. for the isolation of arctiin from *Arctium lappa* L. fruit [49]. The authors considered five extraction parameters, i.e., methanol concentration, microwave power, solid-to-liquid ratio, extraction time, and extraction times. Based on their results, the optimum extraction conditions were 40% methanol, 500 W microwave power, 1:15 solid-to-liquid ratio, 200 s as the extraction time, and three times of the extraction. According to the authors, with these conditions, an extract containing 17.5% of arctiin and 2.2% of arctigenin was obtained, though the efficiency of the extraction is not quite clear as the authors did not include the quantity of plant material used for this experiment. A fine comparison of maceration, Soxhlet extraction, and microwave-assisted extraction was provided by Garg et al. [102]. In their paper, the authors showed that MAE is a superior method for phyllantin extraction to maceration and Soxhlet extraction in terms of lignan yield, shorter extraction time, and lower extraction temperature.

Lastly, supercritical fluid extraction (SFE) may be used as another environmentally friendly extraction technique. In the same way as other alternative methods, it can lower the use of extraction solvents, shorten extraction time, and increase the yield. However, the apparatus needed for the implementation of this technique is much higher than for the previously mentioned methods [24]. This method has been so far mostly used in the extraction of lignans from *Schisandra chinensis* (Turcz.) Baill. fruits, seeds, and leaves. The lignans present in this plant material are free aglycones, with most of them having little to no hydroxyl groups, making them rather lipophilic. Many papers optimizing the *Schisandra* lignans’ SFE extraction were published, with two of them recently [84,85]. Perhaps the most important point from these works is that temperature and pressure variations have a lesser effect on extractability than the addition of methanol or ethanol to carbon dioxide, which dramatically increased the lignan yield [103].

## 6. Artifacts

During the extraction of lignans, artifacts may be formed by unwanted chemical reactions, such as oxidation, polymerization, thermal degradation, and other chemical rearrangements. Their cause usually lies in exposure to light, high temperatures, acidic or alkaline conditions, and the presence of free radicals. Their occurrence is unfortunately rarely reported.

In terms of artifact formation, one of the more sensitive lignans is 7-hydroxymatairesinol. As mentioned before, light irradiation may oxidize 7-hydroxymatairesinol to α-conidendrin, 7-oxomatairesinol, allohydroxymatairesinol, 7-methoxymatairesinol, and vanillin, subsequently forming colored oligomers [31]. Similarly, 7-hydroxymatairesinol may be transformed by acidic and alkaline conditions to α-conidendrin and other oxidation products, which were described in detail by Eklund et al. [104]. Acidic media (extraction solvents, acidic hydrolysis, chromatographic systems, etc.) caused artifact formation from other lignans. Sicilia et al. [34] identified anhydrosecoisolariciresinol, lariciresinol, and cyclolariciresinol in flaxseeds, which were probably products of secoisolariciresinol rearrangements. Other reports confirm the intramolecular cyclization of lariciresinol to cyclolariciresinol [105]. The presence of acetonides and cyclization products of lignans were also reported to occur in some plant substances naturally; thus, it is not always clear whether they should be classified as artifacts [106]. 

Artifact formation due to temperature is relatively rare in lignans’ extraction. As discussed in previous paragraphs, the lignans are generally stable in temperatures up to 100 °C, with some of them even immune to much higher temperatures [27]. Considering that lignans’ extraction, isolation, and purification usually happen at temperatures below this limit, we should not expect degradation products. On the other hand, the temperature effect may enhance the hydrolysis processes and thus the isolation of free aglycones from substances containing only glycosylated forms [28,29]. 

The prevention of artifacts forming can be achieved by implementing simple measures, such as the absence of light, avoidance of high temperatures, control of pH, and storage of extracts in frozen/solid-state [2].

## 7. Isolation and Purification

The separation of individual structures is a crucial step in the chemical analysis of the raw material. The obtained pure compounds can be valuable as new standards for HPLC-MS analysis and for biological experiments to determine their in vitro and in vivo activity.

Flash chromatography on silica gel or Sephadex LH-20 columns has been used for years as the most popular preliminary method for the initial fractionation, and separation, of the samples, as well as for the purification and preparative isolation of natural compounds from raw materials [53]. In addition, various chromatographic methods have been mentioned in the literature, such as open column chromatography [107,108], medium performance liquid chromatography (MPLC) [109], and the most popular method—semi-preparative HPLC [110].

Liquid chromatography is a bidirectional technique, both as an analytical method and as a method of isolating and purifying a single or several compounds from a mixture of many substances, often having very similar molecular structures. Due to the amount of isolated product, preparative chromatography can be divided into several subgroups: semi-preparative chromatography involving the isolation of a few micrograms of analyte (usually for the assessment of initial physiological, toxicological, or pharmacological properties), laboratory-scale preparative chromatography involving the isolation of a few to several grams of analyte as an intermediate in a synthesis process, or for detailed studies of the pharmacological and pharmacokinetic mechanisms under in vivo conditions and process chromatography involving the production of large quantities (many kilograms to tons), commonly for commercialization [111].

Using a semi-preparative HPLC system equipped with a UV–VIS detector is the most versatile method to obtain pure structures from the lignan group with UV absorption monitored at 280 nm or 254 nm. Semi-preparative HPLC analysis can be conducted on a reverse-phase C18 column using an isocratic elution, as was presented during noreucol A, (+)-epiolivil, or (+)-olivil isolation from Eucomiae bark [112], lignans from *Euphorbia hirta* L. [113], and neolignans from *Pouzolzia sanguinea* (Blume) Merr. [114] using a mobile phase containing acetonitrile-H_2_O mixture in various proportions. Similarly, Su et al. performed the isolation of lignans from the seeds of *Arctium lappa* L. obtaining arctigenin, matairesinol, arctiin, and a mixture of two isomers containing lappaol A and isolappaol A, using preparative and semipreparative chromatography [43]. On the other hand, the lignans glucosides from the stems of *Alibertia sessilis* were isolated using the same method but with a methanolic-water mobile phase (30:70) [115]. Trachelogenin, nortrachelogenin, nortracheloside, and other lignans were obtained by Lee et al. from *Trachelospermum asiaticum* Nakai in amounts ranging from 94.6 mg, 25 mg, and 128.2 mg, respectively, using the same mobile phase but with the addition of 0.1% formic acid. The slight addition of acid in the mobile phase is found to be useful for the improvement of peak resolution for the structures containing hydroxyl groups [116]. In contrast, more hydrophilic deoxypodophyllotoxin, with potential anticancer properties, was isolated from *Juniperus communis* L. branches and needles using isocratic, but normal-phase HPLC with an *n*-hexane/EtOAc mixture elution (65:35) [117].

Isolation with preparative HPLC with gradient grade (85: 15 → 0: 100) proved to be an effective method to isolate epipinoresinoil, matairesinol, phylligenin, and arctigenin among the other lignans’ glucosides from *Forsythia ×intermedia* Zabel leaves and flowers (mobile phase consisted of 0.1% HCOOH in H_2_O (A) and 0.1% HCOOH in MeCN (B)) in gram quantities, as was presented by Michalak B et al. [58]. Similarly, gradient elution was used during lignans’ isolation from the seeds of *Carthamus tinctorius* L. The choice of water (A) and methanol (B) as the mobile phase components, at a flow rate of 1 mL/min, ultimately yielded 21.8 mg of trachelogenin, 4.2 mg of arctigenin, and 5.8 mg of matairesinol [53].

Another method used for the separation and purification of components from natural products (including lignans) is centrifugal partition chromatography (CPC), based on continuous liquid–liquid partitioning. This method has many advantages, such as the possibility of using a wide range of solvent systems, less solvent consumption, and an effective way of obtaining a large number of pure target compounds. In addition, this method allows higher mobile phase flow rates to be used, reducing analysis time compared to traditional chromatographic methods. Jeon J. et al. used CPC for the effective purification and isolation of lipid-soluble lignans, such as sesamin and sesamolin from defatted sesame seeds [118]. High-performance countercurrent chromatography (HSCCC) has also been used to purify sesamin and sesamolin from this raw material, but the process yielded product only on the milligram scale [119]. On the other hand, HSCCC proved to be an efficient method for the isolation of secoisolariciresinol diglucoside from *Linum usitatissimum* L., using a gradient method with a yield of 280 mg SDG per 800 mg of flaxseed [73].

## 8. Qualitative and Quantitative Analysis

### 8.1. Thin Layer Chromatography (TLC)

Thin Layer Chromatography has been applied as an inexpensive supporting method in lignan analysis, e.g., during the qualitative examination of plant extracts, fraction collection, monitoring isolation procedures, or the screening of many samples [2]. 

In lignan TLC analysis, silica gel is mostly used, with different eluents and detection methods. The choice of eluent, similar to the extraction solvent consideration, depends on the physicochemical properties of the lignans present in the sample. For instance, Willfor et al. [2] used dichloromethane: ethanol 93:7 (v/v) for samples containing aglycones from conifer wood. Kuehl et al. [53] analyzed samples from seeds of *Carthamus tinctorius* L., containing glucosides of trachelogenin, arctigenin, and matairesinol, with toluene:ethyl acetate:formic acid 3:1:0.2 (v/v/v) and dichloromethane:acetone:formic acid 7:2:0.1 (v/v/v). In another example, Lee et al. [93] studied the flower buds of *Magnolia fargesii* (Finet & Gagnep.) W.C. Cheng, rich in lipophilic lignan aglycones, with *n*-hexane:ethyl acetate 3:1 (v/v). Another example is the chloroform:methanol:water 70:30:4 (v/v/v) solvent system, which was used to analyze the roots of *Eleutherococcus senticosus* (Rupr. & Maxim.) Maxim. and weed of *Viscum album* L. [120].

Various detection techniques have been applied for lignans. As all lignans absorb UV light, plates without chemical treatment can be observed under 254 nm wavelength. Some authors also use 366 nm wavelength to check for the other compounds present in the samples or impurities. Other detection methods include spraying with coloring reagents, such as vanillin in phosphoric or sulfuric acid, and 5% sulfuric acid in ethanol [2,53,93].

Although TLC was previously also used in quantitative analysis (densitometry) as well as for isolation and purification procedures (preparative TLC), it has been nowadays replaced with more advanced methods and its use is rather limited. However, some authors, such as Zare et al. [121], used preparative TLC to purify the lignans isolated from *Linum mucronatum* Bertol. Similarly, Goels et al. [122] isolated pinoresinol from *Picea abies* (L.) H. Karst. balm. One important notion should be highlighted here, that is that the isolation using preparative TLC was executed on a microscale—the yield was 1.2 mg of pinoresinol. 

One of perhaps the newer TLC methods used is thin layer chromatography—direct bio-autophagy (TLC-DB), which allows the direct on-plate identification of active components (mostly regarding antioxidative, antibacterial, and enzyme inhibition properties). Recently, Sobstyl et al. [123] performed the TLC-DB method on *Schisandra chinensis* (Turcz.) Baill. fruit for the analysis of plant metabolites’ acetylcholinesterase inhibition and antibacterial effects.

The High-Performance Thin Layer Chromatography (HPTLC) method was used to quantify sesamin in sesame oil without saponification, which confirms the validity of this method in lignans’ analysis.

### 8.2. High-Performance Liquid Chromatography (HPLC)

High-performance liquid chromatography (HPLC) seems to be the technique of choice applied for the detection, identification, separation, and quantification of lignans from plant matrixes or human fluids. The advantage of this method is, among others, that no complicated sample preparation is required. Based on the literature data, analysis of the lignans was performed by HPLC with various detection methods (Table 2), including UV detection with diode array detector (DAD), coulometric electrode array detection (CEAD), pulse amperometric detection (PAD), fluorescence detector (FLD), and mass spectrometry (MS) which has been considered the most sensitive method for lignan structures to determine their molecular mass and purity. The main advantages of combining liquid chromatography with mass spectrometry include high selectivity, resolution, speed of analyses, sensitivity, repeatability, and the possibility of quantitative analysis with analyte structure determination. In most cases, the analyses are performed with satisfying sensitivity in the UV detection at 280 nm or 254 nm.

Due to the medium polarity of lignans and their metabolites, HPLC is especially suitable for their analysis, on reverse-phase column using a gradient elution mode. The most common reversed-phase column is octadecyl—RP-18 and RP-8—for more hydrophilic structures. In their recent report, Shi et al. also point to the sporadic but successful use of normal-phase liquid chromatography to identify the lipophilic lignans (sesamin, sesamolin, and sesamol) from sesame oil [130] or lignans from the *Podophyllum* species [131]. On the other hand, the *Podophyllum* spp. lignans (podophyllotoxin, deoxypodophyllotoxin, β-peltatin, yatein, matairesinol, anhydropodorhizol) can also be analyzed in the reverse phase system [132].

The most common organic solvents used as the mobile phase in this method are acetonitrile, methanol, and water, with formic, acetic, and phosphoric acids used for pH adjustment. Some modifications, such as mixtures of solvents, e.g., methanol, acetonitrile, and DMSO are necessary for the separation of diastereoisomer combinations (methanol) and functional group derivates (acetonitrile, DMSO). Due to the chiral structure, some lignans occur in plants in enantiomeric forms or enantiomeric excess. HPLC chiral columns with the isocratic flow can be used in this case, both normal and reverse phase ones, with cellulose carbamate as the packing material [131,133].

Dar et al. described the summarized techniques employed for the separation and quantification of lignans (sesamin, sesamolin, secoisolariciresinol, secoisolariciresinol diglucoside, sesaminol triglucoside, pinoresinol, matairesinol) from flaxseed and sesame oil [134]. The main types of chromatography techniques used for quantification in sesame include the many forms of HPLC, such as LC-NMR-MS [72], methods using atmospheric pressure chemical ionization (APCI-MS) [135], and electrospray ionization mass spectrometry (ESI-MS). Hata et al. confirmed the possibility of sesamin detection from leaves using the technique of ultra-performance liquid chromatography–fluorescence detection (UPLC-FLD) [136,137]. In line with literature results, a high-performance liquid chromatography–atmospheric pressure chemical ionization–tandem mass spectrometry (HPLC–APCI–MS/MS) method in comparison to the ESI-MS model appears to be a more generic method in sesame lignans’ analysis, but slightly better for their aglycones than glucosides [136]. In contrast, the combination of an atmospheric pressure-heated nebulizer interface with chemical ionization (HN-APCI) combined with tandem mass spectrometry (HPLC-HN-APCI-MS^n^ (MRM)) was probably the first assay to detect the essential enterolignans (enterodiol, enterolactone) in human serum and urine with a limit of detection in the low parts per billion range [138].

The dibenzocyclooctadiene structures of the lignans from *Schisandra* species were quantitively and qualitatively determined by high-performance liquid chromatography (HPLC) with different detection methods, such as capillary electrophoresis (CE), and mass spectrometry (MS), or by gas chromatography coupled with mass spectrometry (GC-MS). Moreover, due to the strong UV absorbance of the *Schisandra* lignans in the range of 230–255 nm, ultraviolet (UV) detectors can be used as a predominant method with high sensitivity and specificity [139]. Twenty-four lignans, e.g., schisanhenol, schisandrin A-C, and gomisin A-O, were also confirmed with the UHPLC–MS/MS method from fruits and leaves of *Schisandra rubiflora* Rehder & E.H. Wilson [128].

Another method allowing the simultaneous separation and structural identification of compounds in the mixtures or plant extracts is liquid chromatography coupled with nuclear magnetic resonance (LC-NMR). It is one of the most powerful means of compound identification, which solved problems with signal interferences by the eluents and sensitivity by solid-phase extraction cartridge (SPE) application. The reversed-phase HPLC-SPE-NMR method has found application in the identification of structurally similar lignans from *Phyllanthus urinaria* L. [140]. The use of LC-NMR coupled with mass spectrometer (LC-NMR-MS) was also noted in lignan analysis, for example in the structural elucidation of the isolated and purified secoisolariciresinol diglucoside diastereomers obtained from flaxseed [72].

### 8.3. Liquid Chromatography Mass Spectrometry

The development of hyphenated techniques related to liquid chromatography and electrospray ionization mass spectrometry (LC-ESI-MS) has provided a routine method for the identification of plant metabolites (including lignans) in complex plant matrixes. In most cases, the compounds are identified by comparing fragmentation patterns with available libraries or using reference compounds [141,142]. A valuable work on the identification of lignans by liquid chromatography–electrospray ionization ion-trap mass spectrometry was completed by Eklund et al. [141]. Table 3 presents the fragmentation data of the major lignans based on gathered reports. 

The most common ionization methods used during MS/MS analysis of lignans include negative electrospray ionization (ESI), negative atmospheric pressure chemical ionization (APCI), and negative ionization in a heated nebulizer (HN).

LC-ESI-MS^n^ (ion trap) method is a method of choice, because of its shorter run time, smaller injection, high sensitivity, and selectivity. This method is useful for detecting structures with multiple ingredients because of the lower limit of detection and ability to elucidate their structure. LC-ESI-MS can be recorded in the positive and negative ion mode, with the latter being used in most cases [153]. This is because the acid phenolic groups found in the lignan structures, have a good proton-donating capacity, and thus are easily deprotonated (e.g., secoisolariciresinol, enterodiol), apart from lignans with methylenedioxy-bridged furofuran structures which, due to the lack of phenolic hydroxyl groups, can be optimally detected in the positive ionization mode [136]. Deoxypodophyllotoxin and its precursors from *Anthriscus sylvestris* (L.) Hoffm. roots were analyzed using this method [154], as well as honokiol and magnolol from *Magnolia officinalis* Rehder & E.H. Wilson [155]. In contrast, a QTRAP LC-MS system with hybrid triple quadrupole/linear ion trap mass spectrometer has been successfully applied by Cui et al. to analyze bioactive constituents (including lignans) in different parts of *Forsythia suspensa* Vahl [144].

As a method of high specificity, HPLC-ESI-MS happens to be useful for the determination in the negative-ion scan mode of flaxseed lignans (secoisolariciresinol, secoisolariciresinol diglucoside) and enterolignans (enterodiol, enterolactone, and their glucuronides) from urine and blood samples, 12 and 24 h after lignan intake [156]. On the other hand, 11 lignans from *Magnolia biondii* Pamp. (furofuran structure) were identified with the same method using positive ion mode [126]. LC-ESI-MS also allowed for the differentiation of the metabolite profiles, including lignans (i.e., schisandrin, deoxyschisandrin, and schisandrin B), between two species: *Schisandra chinensis* (Turcz.) Baill. and *Schisandra sphenanthera* Rehder & E.H. Wilson [157]. Phyllanthin and related lignans were also analyzed by UPLC-ESI-MS, using 0.1% formic acid in methanol and 0.1% formic acid in water as the mobile phases. Five lignans (two new, and three known structures) isolated from *Zanthoxylum armatum* DC peels, as well as another five lignans (euphorhirtins A-D, 5-methoxyvirgatusin) from *Euphorbia hirta* L. were analyzed using high resolution mass spectrometry (HRMS). Two dibenzocyclooctadiene lignans, i.e., schisphenlignan M and N, from ethanol extract of the root bark of *Schisandra sphenanthera* Rehder & E.H. Wilson [158] and lignans from *Ginkgo biloba* L. roots were analyzed using the same method [159].

HPLC-TOF-MS and HPLC-DAD-ESI-QTOF-MS analysis allowed the identification of nudiposide, a lignan from avocado (*Persea americana* Mill.) seed and seed coat [160], as well as lignans (arctiin, arctigenin, matairesinol) and sesquilignans (lappaol C, isolappaol C, lappaol A, isolappaol A) from *Cnicus benedictus* L. fruit samples [161], and olivil-type lignans among other types from *Eucommia ulmoides* Oliv. bark [112]. UPLC-ESI-QTOF-MS proved to be a good option for confirming the presence of dibenzylbutyrolactone lignans from Great Burdock seeds (*Arctium lappa* L.) [162] and furofuran-type lignans (pinoresinol, epipinoresinol) from *Carduus nutans* L. fruits [163]. Doussot et al. for the first time successfully analyzed the content of lignans from wild flax species, such as *Linum flavum* L., as well as *Juniperus* and *Callitris* species for the content of lignan podophyllotoxin and its precursors by liquid chromatography coupled with high-resolution mass spectrometer (LC-HRMS-QTOF-MS) [145].

### 8.4. Gas Chromatography Mass Spectrometry

Gas chromatography (GC) is a less popular method for the identification of phenolic compounds from the lignan group due to their low volatility and the need for high temperatures, which can damage the analytes [164]. On the other hand, this method allows the identification of multicomponent mixtures (e.g., polyphenols), combined with a detection system (usually a mass spectrometer—GC-MS) giving unambiguous information about its composition, and thus can be an alternative to LC-MS.

Although gas chromatography coupled with mass spectrometry (GC-MS) is far from being a method of choice for lignans’ analyses, the attempts to identify some of them (e.g., pinoresinol, matairesinol, sesamin, asarinin, sesamolin, medioresinol secoisolariciresinol, isolariciresinol, anhydrosecoisolariciresinol, lariciresinol, isolariciresinol, and syringaresinol among others) by GC-MS can be found in the literature [165,166,167,168,169]. 

The biggest obstacle in non-volatile lignans’ analysis by GC-MS is the need for sample derivatization to make the functional groups (i.e., hydroxyl groups) detectable. Furthermore, the derivatization process guarantees an increase in the thermal stability of the analytes to prevent their degradation under analytical conditions, thereby increasing the sensitivity and specificity of the assay. The most common techniques for the derivatization of lignan compounds include silylation and acylation or combinations of both methods. As an example of the reagents used to derivatize labile hydroxyl groups, for example in extracts from pomegranate by-products [168] and juices or flaxseed extracts [166], is BSTFA reagent (N,O-Bis(trimethylsilyl)trifluoroacetamide), which among others requires the longest time for the derivatization process [166]. Other common silylation mixtures include pyridine with BSTFA containing 1% trimethylchlorosilane (TMCS) [166], hexamethyldisilazane/trimethylchlorosilane in pyridine 2:1:10 (Tri-Sil reagent), N,O-bis(trimethylsilyl)acetamide (BSA), deuterated BSA [170], or pentafluoropropionic anhydride (PFPA) [171]. During the pinoresinol, matairesinol, and secoisolariciresinol analysis in various food samples, a Tri-Sil reagent was used in the derivatization process [165]. Some specific lignans, e.g., dibenzocyclooctadiene type from the fruit of the *Schisandrae* spp. and *podophyllotoxin*-type lignans, can be detected directly without the need for derivatization because of the lack of hydroxyl groups [172,173].

The accurate quantification of the key human lignan metabolites, such as enterodiol and enterolactone, is drawing attention due to their beneficial protective effects in various diseases. The successful detection of enterolignans in human plasma has been accomplished primarily using liquid chromatography–mass spectrometry (LC-MS) or gas chromatography–mass spectrometry (GC-MS) operated in full scan mode [174,175]. Edel et al. proposed a successful method for the precise quantification of enterolignans after consumption of flaxseed using supported liquid extraction (SLE) with quantification using gas chromatography and mass spectrometry in the micro-selected ion storage mode (GC-MS-μSIS) [176].

During GC-MS lignans’ analyses, helium is commonly used as the carrier gas at a constant flow rate of 1 mL/min [165,176]. According to the literature data, analyses are carried out with a gradual increase in the detector temperature from 270–280 °C [165,166,176], up to 320–340 °C [175], in total ion current (TIC) mode and/or selected ion monitoring (SIM) mode. The most common choice of a chromatography column is standard capillary non-polar columns (DB-17, HP-1) [177] or columns with mid polarity i.e., HP-5 [166], DB-17 [172], and VF-5 [176]. The electron impact (EI) is the most frequently chosen ionization technique in the GC-MS analysis of lignans and enterolignans [176].

An interesting addition to the methods of identifying the presence of specific compounds in the plant is the direct visualization of their distribution in individual parts of the raw material. Yu et al. first performed visualization of lignans’ and neolignans’ compounds (olivil, pinoresinol, lariciresinol, secoisolariciresinol, and their glucosides) in *Gingko* spp. stems using cryo-temporal secondary ion mass spectrometry coupled with scanning electron microscopy (cryo-TOF-SIMS/SEM). This method appears to be an essential element necessary to understand their biological functions by allowing visualization of their distribution with submicron transverse resolution in two-dimensional images [178].

## 9. Conclusions

This review provided a summary of both the conventional and alternative methods used to extract lignans from plant material. As isolation of pure compounds may be time-consuming, complicated, expensive, and impact the environment, an appropriate technique should be selected. Among the presented methods, alternative or advanced methods such as UAE, MAE, and SFE are rapid, efficient, consume fewer extraction solvents, and are environmentally friendly. Analytical methods for detecting lignans have also been discussed, especially emphasizing the advantages of hyphenated methods, such as LC-MS and GC-MS. Since most lignans have free hydroxyl groups, LC-MS should be regarded as a superior technique in most cases. Nevertheless, TLC and other conventional techniques still have a role as auxiliary methods in lignans’ analysis. 

In summary, many previous reports have shown that lignans may interfere with cancer cells or even offer a means of preventing carcinogenic diseases. These compounds are present in the everyday human diet and thus should be more comprehensively studied to develop new therapies. A properly selected method for extracting and isolating lignans from the plant material may facilitate this process.

## Figures and Tables

**Figure 1 plants-11-02323-f001:**
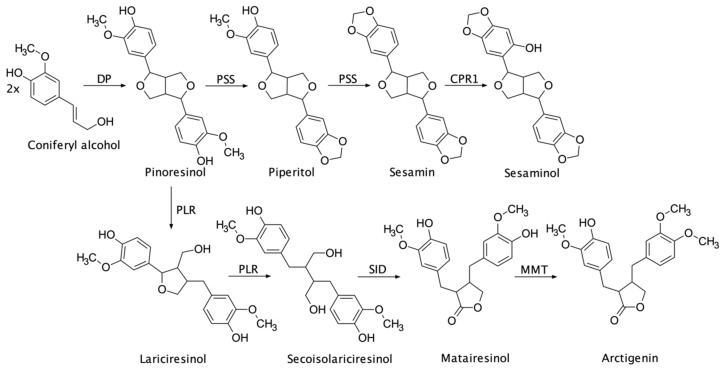
Biosynthetic pathway of major lignans reconstructed from the published records [1,4,5,6,7,8,9,10,11]. DP (dirigent protein), PSS (piperitol/sesamin Synthase), PLR (pinoresinol/lariciresinol reductase), CPR1 (cytochrome P450 oxidoreductase 1), SID (secoisolariciresinol dehydrogenase), MMT (matairesinol O-methyltransferase).

**Table 1 plants-11-02323-t001:** Comparison of methods used in lignans’ extraction.

Species	Plant Part	Method	Solvent	Extraction Times	Extraction Time (min)	Temperature (°C)	Solvent to Sample Ratio	Reference
*Abies alba* Mill.	bark	Digestion	H_2_O	1×	120	70	1:5	[39]
*Abies chensiensis* Tiegh.	aerial parts	Maceration	80% EtOH/H_2_O (v/v)	3×	180	RT	-	[46]
*Abies delavayi var. nukiangensis* (W.C.Cheng & L.K.Fu) Farjon & Silba	aerial parts	Maceration	95% MeOH/H_2_O (v/v)	3×	180	RT	-	[47]
*Arctium lappa* L.	fructus	Digestion	95% EtOH/H_2_O (v/v)	2×	180	80	1:10	[48]
*Arctium lappa* L.	fructus	MAE	40% MeOH/H_2_O (v/v)	3×	200 s	-	1:15	[49]
*Arctium lappa* L.	fructus	Soxhlet/heated reflux	70% EtOH/H_2_O (v/v)	3×	60	BP	1:8	[50]
*Arctium lappa* L.	fructus	Soxhlet/heated reflux	80% EtOH/H_2_O (v/v)	3×	-	BP	1:6	[51]
*Arctium lappa* L.	fructus	UAE	MeOH	1×	20	-	1:90	[42]
*Arctium lappa* L.	fructus	UAE	MeOH + H_2_O	1×	30 (with MeOH) and 30 (with water)	-	1:15	[29]
*Arctium lappa* L.	roots	Maceration	80% MeOH/H_2_O (v/v)	1×	8 h	RT	1:50	[25]
*Arctium lappa* L.	seed	Maceration	MeOH	3×	-	RT	1:2	[43]
*Arctium lappa* L.	seed	Maceration	MeOH	3×	-	RT	3:5	[52]
*Arctium lappa* L.	seeds	Maceration	80% MeOH/H_2_O (v/v)	1×	8 h	RT	1:50	[25]
*Carthamus tinctorius* L.	seed	Maceration	MeOH	5×	15 h	RT	1:2	[53]
*Eleutherococcus divaricatus* (Siebold & Zucc.) S.Y.Hu	root	Soxhlet/heated reflux	MeOH	3×	-	BP	1:2	[54]
*Eleutherococcus senticosus* (Rupr. & Maxim.) Maxim.	bark	Soxhlet/heated reflux	70% MeOH/H_2_O (v/v)	2×	180	BP	1:6	[55]
*Eleutherococcus senticosus* (Rupr. & Maxim.) Maxim.	root	Soxhlet/heated reflux	H_2_O	3×	30	BP	1:6	[37]
*Eleutherococcus senticosus* (Rupr. & Maxim.) Maxim.	stem	Maceration	MeOH	1×	2 weeks	RT	-	[56]
*Eleutherococcus sessiliflorus* (Rupr. & Maxim.) S.Y.Hu	fruit	Maceration	70% EtOH/H_2_O (v/v)	3×	24 h	RT	5:18	[57]
*Forsythia × intermedia* Zabel	flower	Digestion	75% MeOH/H_2_O (v/v)	6×	120	70	1:20	[58]
*Forsythia × intermedia* Zabel	flower	Soxhlet/heated reflux	MeOH	4×	-	BP	1:200	[44]
*Forsythia × intermedia* Zabel	leaf	Digestion	75% MeOH/H_2_O (v/v)	6×	120	70	1:20	[58]
*Forsythia × intermedia* Zabel	leaf	Soxhlet/heated reflux	MeOH	4×	-	BP	1:200	[44]
*Forsythia koreana* (Rehder) Nakai	flower	Maceration	80% MeOH/H_2_O (v/v)	2×	24 h	RT	3:100	[59]
*Forsythia koreana* (Rehder) Nakai	fruit	Maceration	MeOH	3×	7 days	RT	-	[60]
*Forsythia koreana* (Rehder) Nakai	fruit	Maceration	MeOH	3×	-	RT	1:1	[61]
*Forsythia koreana* (Rehder) Nakai	stem	UAE	MeOH	1×	60	-	1:2	[62]
*Forsythia suspensa*	flower	Soxhlet/heated reflux	MeOH	4×	-	BP	1:200	[44]
*Forsythia suspensa*	fruit	Maceration	70% EtOH/H_2_O (v/v)	4×	-	RT	2:1	[63]
*Forsythia suspensa*	fruit	Soxhlet/heated reflux	50% MeOH/H_2_O (v/v)	1×	60	BP	1:25	[64]
*Forsythia suspensa* (Thunb.) Vahl	fruit	Soxhlet/heated reflux	60% EtOH/H_2_O (v/v)	2×	120	BP	-	[65]
*Forsythia suspensa* (Thunb.) Vahl	fruit	Soxhlet/heated reflux	95% EtOH/H_2_O (v/v)	3×	120	BP	1:60	[66]
*Forsythia suspensa* (Thunb.) Vahl	fruit	Soxhlet/heated reflux	MeOH	7×	-	BP	3:10	[45]
*Forsythia suspensa* (Thunb.) Vahl	fruit	UAE	20% MeOH/H_2_O (v/v)	1×	30	-	1:5	[67]
*Forsythia suspensa* (Thunb.) Vahl	leaf	Soxhlet/heated reflux	MeOH	4×	-	BP	1:200	[44]
*Forsythia viridissima* Lindl.	flower	Soxhlet/heated reflux	MeOH	4×	-	BP	1:200	[44]
*Forsythia viridissima* Lindl.	leaf	Soxhlet/heated reflux	MeOH	4×	-	BP	1:200	[44]
*Forsythia viridissima* Lindl.	root	UAE	80% MeOH/H_2_O (v/v)	3×	90	RT	7:10	[68]
*Larix laricina* (Du Roi) K.Koch	bark	Maceration	80% EtOH/H_2_O (v/v)	1×	24 h	RT	1:10	[69]
*Linum usitatissimum* L.	seed	Maceration	60% EtOH/H_2_O (v/v)	2×	60	RT	10:75	[70]
*Linum usitatissimum* L.	seed	MAE	70% MeOH/H_2_O (v/v)	1×	180	60	25:1	[71]
*Linum usitatissimum* L.	seed	Digestion	75% MeOH/H_2_O (v/v)	1×	24 h	65	-	[72]
*Linum usitatissimum* L.	seed	Digestion	80% EtOH/H_2_O (v/v)	1×	240	55	1:14	[34]
*Linum usitatissimum* L.	seed	Maceration	70% MeOH/H_2_O (v/v)	1×	120	RT	1:3	[73]
*Linum usitatissimum* L.	seed	Maceration	H_2_O	1×	60	RT	1:15	[70]
*Linum usitatissimum* L.	seed	UAE	70% MeOH/H_2_O (v/v)	3×	4h	-	1:6	[74]
*Linum usitatissimum* L.	seed	UAE	H_2_O + 0.2 N NaOH	1×	60	25	-	[75]
*Linum usitatissimum* L.	seed	UAE	50% MeOH/H_2_O (v/v)	1×	10 + 60	RT	1:25	[76]
			70% acetone/H_2_O (v/v)	1×	-	RT	1:25	
*Magnolia biondii* Pamp.	flower buds	Soxhlet/heated reflux	MeOH	1×	5 h	BP	-	[77]
*Magnolia officinalis* Rehder & E.H. Wilson	flower buds	Soxhlet/heated reflux	MeOH	3×	180	BP	9:4	[78]
*Pinus radiata* D.Don	bark	Soxhlet/heated reflux	70% acetone/H_2_O (v/v)	4×	180	BP	1:10	[79]
*Pinus sylvestris* L.	wood	ASE	95% acetone/H_2_O (v/v)	2×	5	100	-	[80]
*Pinus sylvestris* L.	wood	Soxhlet/heated reflux	90% acetone/H_2_O (v/v)	1×	180	BP	3:250	[81]
*Saraca asoca* (Roxb.) Willd.	bark	MAE	70% MeOH/H2O (v/v)	1×	10	-	1:30	[82]
*Schisandra chinensis* (Turcz.) Baill.	fruit	Digestion	70% MeOH/H_2_O (v/v)	2×	5 h	50	-	[83]
*Schisandra chinensis* (Turcz.) Baill.	fruit	SFE	CO_2_ + MeOH	2×	20	-	1:10	[84]
*Schisandra chinensis* (Turcz.) Baill.	wood	SFE	CO_2_ + EtOH	1×	6 h	-	-	[85]
*Sesamum indicum* L.	seed	Maceration	80% EtOH/H_2_O (v/v)	2×	8 h	RT	1:10	[86]
*Sesamum indicum* L.	seed	Maceration	cyclohexane + DCM + MeOH (1:1:1)	3×	-	RT	1:1	[36]
*Sesamum indicum* L.	seed	Maceration	H_2_O	1×	24 h	RT	-	[40]
*Sesamum indicum* L.	seed	Soxhlet/heated reflux	*n*-hexane	1×	10 h	BP	-	[26]
*Sesamum indicum* L.	seed	Soxhlet/heated reflux	MeOH	5×	8 h	BP	1:10	[87]
*Sesamum indicum* L.	seed	UAE	50% MeOH/H_2_O (v/v)	1×	10 + 60	RT	1:25	[88]
			70% acetone/ H_2_O (v/v)	1×	-	RT	1:25	
*Sesamum indicum* L.	seed	UAE	80% acetone/H_2_O (v/v)	2×	5	-	1:10	[30]
*Syringa pinnatifolia* Hemsl.	bark	Soxhlet/heated reflux	95% EtOH + 80% EtOH	2×	90	BP	1:2	[89]
*Syringa pinnatifolia* Hemsl.	root	Maceration	95% EtOH/H_2_O (v/v)	3×	-	RT	1:5	[90]
*Syringa vulgaris* L.	bark	Digestion	60% EtOH/H_2_O (v/v)	1×	60	70	1:20	[91]

ASE—accelerated solvent extraction; BP—boiling point; DCM—dichloromethane; EtOH—ethanol; MAE—microwave-assisted extraction; MeOH—methanol; RT—room temperature; SFE—supercritical fluid extraction; UAE—ultrasound-assisted extraction.

**Table 2 plants-11-02323-t002:** HPLC methods used in the analysis of lignans.

Species	Plant Part	Mobile Phase	Column	Elution Type	Detection System	Reference
*Abies alba* Mill.	bark	H_2_O and MeCN	RP-C_18_ column (10 cm × 4.6 mm, 2.7 µm)	gradient	DAD	[39]
*Arctium lappa* L.	fruit	H_2_O + 0.01% HCOOH and MeCN	RP-C_18_ (250 mm × 4.6 mm, 5 µm)	gradient	DAD	[28]
*Arctium lappa* L.	fruit	H_2_O and MeCN	RP-C_18_ (150 mm × 4.6 mm, 5 μm)	gradient	FLD	[42]
*Arctium lappa* L.	fruit	MeCN and H_2_O + 0.1% HCOOH	RP-C_18_ (250 mm × 4.6 mm, 5 μm)	gradient	UV detection at 254 nm	[29]
*Arctium lappa* L.	fruit, root	H_2_O + 0.1% HCOOH and MeOH + 0.1% HCOOH	RP-C_18_ (250 mm × 2.0 mm, 5 µm)	gradient	DAD	[25]
*Carthamus tinctorius* L.	fruit	H_2_O + MeOH	RP-C_12_ (150 mm × 4.6 mm, 3.5 µm)	gradient	UV detection at 221/ESI-MS	[53]
*Eleutherococcus senticosus* (Rupr. & Maxim.) Maxim.	bark	H_2_O + 0.2% CH_3_COOH and MeOH + 0.2% CH_3_COOH	RP-C_18_ (125 mm × 3 mm, 3 µm)	gradient	UV detection at 210 nm, 254 nm and 280 nm	[55]
*Eleutherococcus senticosus* (Rupr. & Maxim.) Maxim.	root	H_2_O + 15% MeCN + 0.1% HCOOH	RP-C_18_ (250 mm × 4.6 mm, 5 µm)	isocratic	UV detection at 205 nm	[37]
*Eucommia ulmoides* Oliv.	bark	H_2_O + 0.1% HCOOH and MeOH	RP-C_18_ (50 mm × 2.1 mm, 1.7 μm)	gradient	ESI-MS	[124]
*Forsythia × intermedia* Zabel	flower, leaf	H_2_O + 0.1% HCOOH and MeCN + 0.1% HCOOH	RP-C_18_ (150 mm × 2.1 mm, 1.9 μm)	gradient	DAD, ESI-MS	[58]
*Forsythia koreana* (Rehder) Nakai	stem	H_2_O + 32% MeCN	RP-C_18_ (150 mm × 4.6 mm, 5 μm)	isocratic	UV detection at 254 nm	[62]
*Forsythia suspensa* (Thunb.) Vahl	fruit	25% MeCN + 0.1% HCOOH	RP-C_18_ (250 mm × 4.6 mm, 5 μm)	isocratic	UV detection at 250 nm	[65]
*Forsythia suspensa* (Thunb.) Vahl	fruit	H_2_O + 0.1% HCOOH and MeOH	RP-C_18_ (150 mm × 4.6 mm, 5 μm) coupled with RP-C18 (12.5 mm × 4.6 mm, 5 μm)	gradient	DAD	[64]
*Linum usitatissimum* L.	seed	H_2_O + 0.2% CH_3_COOH and MeCN	RP-C_18_ (250 mm × 4.6 mm, 5 µm)	gradient	UV detection at 280 nm	[73]
*Linum usitatissimum* L.	seed	H_2_O + 0.2% CH_3_COOH and MeCN	RP-C_18_ (250 mm × 4.6 mm, 5 µm)	gradient	UV detection at 280 nm	[71]
*Linum usitatissimum* L.	seed	H_2_O + 0.2% CH_3_COOH and MeOH	RP-C_18_ (250 mm × 4.0 mm, 5 µm)	gradient	DAD	[75]
*Linum usitatissimum* L.	seed	H_2_O + 16% MeOH + HCOOH and 100% MeOH	RP-C_18_ (250 mm × 4.6 mm, 5 µm)	gradient	UV detection at 283 nm	[34]
*Linum usitatissimum* L.	seed	H_2_O and MeCN + HCOOH	RP-C8 (250 mm × 4 mm, 5 µm)	isocratic	CEAD	[125]
*Magnolia biondii* Pamp.	flower	H_2_O + 0.2% HCOOH and MeCN	RP-C_18_ (250 mm x 4.6 mm, 5 μm)	gradient	UV detection at 278 nm/ESI-MS	[126]
*Schisandra chinensis* (Turcz.) Baill.	fruit	H_2_O + 0.1% H_3_PO_4_ and MeCN	RP-C_18_ (150 mm × 2.0 mm, 5 µm)	gradient	UV detection at 280 nm	[83]
*Schisandra chinensis* (Turcz.) Baill.	fruit	H_2_O and MeCN	RP-C_18_ (250 mm × 4.6 mm, 5 µm)	gradient	UV detection at 215 nm	[127]
*Schisandra rubriflora* Rehder & E.H.Wilson	fruit, leaf	H_2_O and MeOH + 0.1% HCOOH	RP-C_18_ (150 mm × 4.6 mm, 2.7 µm)	gradient	ESI-MS	[128]
*Schisandra sphenanthera* Rehder & E.H.Wilson	fruit	H_2_O and MeCN	RP-C_18_ (250 mm × 4.6 mm, 5 µm)	gradient	PAD-MS, ESI-MS	[129]
*Sesamum indicum* L.	seed	H_2_O + 0.5% CH_3_COOH and MeCN	RP-C_18_ (150 mm × 4.6 mm, 2.7 µm)	gradient	DAD	[88]
*Sesamum indicum* L.	seed	H_2_O and MeOH	RP-C_18_ (250 mm × 4 mm, 10 µm)	gradient	UV detection at 280 nm	[40]
*Sesamum indicum* L.	seed	H_2_O + 0.5% CH_3_COOH and MeCN	RP-C_18_ (150 mm × 2.1 mm, 1.9 µm)	gradient	APCI-MS, ESI-MS	[130]
*Syringa vulgaris* L.	bark	H_2_O + 0.1% HCOOH and MeCN + 0.1% HCOOH	RP-C_18_ (150 mm × 2.1 mm, 1.9 µm)	gradient	DAD, ESI-MS	[91]

APCI-MS—atmospheric pressure chemical ionization mass spectrometry; ESI-MS—electrospray ionization mass spectrometry; CEAD—coulometric electrode array detection; DAD—diode array detector; FLD—fluorescence detection; MeCN—acetonitrile; PAD-MS—photodiode array detection mass spectrometry; RP—reversed phase.

**Table 3 plants-11-02323-t003:** MS fragmentation patterns of major lignans.

Compound	ESI ^a^	Extracted Ion [m/z]	Fragment Ions [m/z]	Reference
Arctigenin	[M−H]^−^	371	356, 295, 209	[58]
Arctigenin glucoside	[M+HCOO]^−^	579	371	[58]
	[M−H]^−^	533	371	
Aschantin	[M+H]^+^	401	365, 353, 261, 231, 219, 181, 151	[126]
	[M+H−H_2_O]^+^	383		
Conidendrin	[M−H]^−^	355	340, 286, 147	[143]
Cyclolariciresinol	[M−H]^−^	359	344, 329, 313, 159	[141]
Epipinoresinol	[M−H]^−^	357	151	[144]
Epipinoresinol glucoside	[M+HCOO]^−^	565	357	[145]
	[M−H]^−^	519	357	
	[M−H−Glc]^−^	357	-	
Fargesin	[M+H]^+^	371	335, 323, 283, 231, 219, 151	[126]
	[M+H−H_2_O]^+^	353		
Hinokinin	[M+H]^+^	355	337, 319, 261, 135	[141]
7-Hydroxylariciresinol	[M+Na^+^]^+^	399	384, 381, 369, 351, 219, 202	[146]
7-Hydroxymatairesinol	[M−H]^−^	373	355, 340, 311, 296, 231, 160	[141]
Lariciresinol	[M−H]^−^	359	344, 329, 208, 161	[143]
Magnolin	[M+H]^+^	417	381, 369, 329, 279, 249, 231, 219, 189	[126]
	[M+H−H_2_O]^+^	399		
Matairesinol	[M−H]^−^	357	342, 313, 298, 281, 209	[58]
Matairesinol glucoside	[M−H]^−^	519	357	[58]
	[M−H−Glc]^−^	357	342, 313, 298, 281, 209	
Medioresinol	[M−H]^−^	387	372, 181, 166, 151, 123	[141]
Medioresinol diglucoside	[M−H]^−^	711	548, 387	[124]
Nortrachelogenin	[M−H]^−^	373	355, 327, 235, 223	[141]
Olivil	[M−H]^−^	375	357, 345, 327, 195, 179, 164	[147]
Olivil glucoside	[M−H]^−^	537	375, 345, 327, 195, 179	[124]
	[M−H−Glc]^−^	375	-	
Olivil diglucoside	[M+HCOO]^−^	745	-	[124]
	[M−H]^−^	699	375, 345, 195, 179	
	[M−H−Glc]^−^	537	327	
7-Oxomatairesinol	[M−H]^−^	371	356, 327, 205	[141]
Phillygenin	[M−H]^−^	371	356	[58]
Phillygenin glucoside	[M+HCOO]^−^	579	371	[58]
	[M−H]^−^	533	371	
Pinoresinol	[M−H]^−^	357	342, 311, 151, 136	[141]
Pinoresinol glucoside	[M−H]^−^	519	357	[145]
	[M−H−Glc]^−^	357	342, 311, 151, 136	
Pinoresinol diglucoside	[M+HCOO]^−^	727	-	[124]
	[M−H]^−^	681	357, 151	
	[M−H−Glc]^−^	519	-	
Schizandrin	[M+H]^+^	433	415, 384, 369	[148]
Secoisolariciresinol	[M−H]^−^	361	346, 331, 313, 179, 165	[141]
Secoisolariciresinol diglucoside	[M+2Na^+^−H]^−^	732	722, 686	[149]
Sesamin	[M+H]^+^	355	353, 337	[150]
	[M+H−H_2_O]^+^	337	319, 289, 261, 231, 203	
	[M+H−H_2_]^+^	353	323, 135, 77	
Sesaminol	[M−H]^−^	369	-	[151]
Sesaminol glucoside	[M−H]^−^	531	-	[151]
Sesaminol diglucoside	[M−H]^−^	693	-	[151]
Sesaminol triglucoside	[M−H]^−^	855	-	[151]
Syringaresinol	[M−H]^−^	417	402, 181, 166, 151	[141]
Syringaresinol glucoside	[M−H]^−^	579	417, 181	[124]
	[M−H−Glc]^−^	417	-	
Syringaresinol diglucoside	[M−H]^−^	741	579, 417, 181	[124]
	[M+HCOO]^−^	787	579	
	[M−H−Glc]^−^	579	-	
Trachelogenin	[M−H]^−^	387	357, 339, 329, 249, 193	[141]
Trachelogenin glucoside	[M+Na^+^]^+^	573	389, 371, 343, 325, 247, 203, 151, 137	[152]

^a^ cation or anion formed and type of ionization mode (positive or negative); [M−H]^−^—deprotonated molecule; [M+H]^+^—protonated molecule; [M+HCOO]^−^—formate adduct ion; [M−H−Glc]^−^—deprotonated and deglycosylated molecule; [M+Na^+^]^+^—sodium adduct ion.

## Data Availability

Not applicable.

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
