# Peer review of "Extraction Techniques and Analytical Methods for Isolation and Characterization of Lignans"

_plants, 2022, doi:10.3390/plants11172323_

Round 1

Reviewer 1 Report

Dear Authors

1. Indicate how the information was selected (articles?, books?, patents?, congresses?, etc). In which databases did you consult and download the information? what was the search key you used? The collection of information was carried out between which dates?

2. It write the requested information in the manuscript.

Regards

Reviewer

Author Response

Dear Reviewer,

thank you for your suggestions. The information you requested (Points 1 and 2) was added to the manuscript in the version presented below:

Literature search for the present review was conducted using the following databases: PubMed, Scopus, and Google Scholar with the following search terms: “lignans”, “extraction”, “isolation”, “detection”, “quantification” and “analysis” in various combinations. Results were first screened for their relevance based on their abstract and afterwards full texts were analyzed. Studies from the year 1998 – 2022 were considered, as this period was characterized by considerable interest in lignan analysis. The search was conducted on July 4th, 2022.

Reviewer 2 Report

I have carefully reviewed the review titled Extraction Techniques and Analytical Methods for the Isolation and Characterization of Lignans. I am satisfied with the research and review work carried out, important topics are observed in the extraction and purification of lignans, it is also noteworthy that there is talk of more friendly methods with the environment (green extraction methods). However, there are some minor writing spots that need to be corrected.

-       In line 20 the name of the botanical species must be written in italics “Podophyllum peltatum”. The same in the lines 86 and 295. 

-       in line a52 the reference is missing, correct it. “….stereoselective biosynthesis (Error! Reference source not found.)”

-       Line 412 biological experiments to determine their in vitro and in vivo activity. in vitro and in vivo must be written in italics

-       In Line 442 (Blume) …containing acetonitrile-H2O. H2O must be written in italics. the same for the line 464 .

It is suggested to the authors to add a column in table 1, which indicates, if available in the reviewed bibliography, the percentages of obtaining extracts by the different techniques and mixtures of solvents used. This way it would be more informative and useful to readers.

Author Response

Dear Reviewer,

thank you for your suggestions.

All indicated errors in style have been corrected.

Regarding your suggestion:

It is suggested to the authors to add a column in table 1, which indicates, if available in the reviewed bibliography, the percentages of obtaining extracts by the different techniques and mixtures of solvents used. This way it would be more informative and useful to readers.

Although, as you have pointed out this information would be very informative and useful to readers such column would be empty in most places due to lack of information. Only a few papers indicated exact mass of dry extract after extraction with the information on the amount of plant material used, which would allow us to calculate the percentage of obtained extract. In many cases mass of the extract is indicated after subsequent steps (e.g. partitioning with ethyl acetate, deffating) or even after column chromatography. Sometimes the information is presented in ml rather than in grams. Due to these many issues, we don't think that such a column would present information on the extract yield accuretly.

Reviewer 3 Report

Dear Authors,

This review addresses different extraction and isolation techniques of lignans. The review is well-structured, written in a good manner and highly interesting to read. The tables are summarizing the information in such an efficient way. The overall manuscript is a collective and interesting piece of knowledge of lignans.
Nevertheless, there are some notes to be considered. 

Abstarct

14

Lignans

Lignans (not in bold format)

20

Podophyllum peltatum.

Podophyllum peltatum.

(in italic format)

25

In vitro and in vivo

In vitro, in vivo

(in italic format)

Introduction

52

No reference found kindly add

86

Phyllanthus

Phyllanthus

(in italic format)

89

Same note for in vitro and in vivo

Solvents

187, 195

glucosides

glycosides (more accurate)

In all manuscript

n-hexane

n-hexane

Solvents

In the table “Comparison of methods used in lignans extraction”

Regarding the reference (87, Mekky et al 2019) please correct the used solvent system as it was a dual extraction method. The first step was with defatting with n-hexane followed by 50% MeOH:H2O followed by sonication, then the second step was with 70% Acetone:H2O. This method afforded extraction of lignans without hydrolysis of their glycosides. Kindly check that there are up to four hexosides conjugates in some observed lignans of sesame seeds cake.

Similarly, it is advisable to add the reference regarding linseeds cake, where a similar method was used and afforded the first detection of Secoisolariciresinol trihexoside dihydroxymethyl glutaryl ester without the hydrolysis in any of secoisolariciresinol conjuagtes.

https://doi.org/10.1016/j.foodchem.2022.133524

Methods

Table 3 please be sure to use the Font of MDPI “Palatino Linotype”

Author Response

Dear Reviewer,

thank you for your suggestions.

All indicated errors in style have been corrected and missing reference (to the figure 1) was added. 

The term "glucoside" was indeed a spelling error, as we wanted to refer to all glycosides.

Regarding reference 87, we have made changes to the table adding omitted second step, though we won't be adding information about deffating in the table (as this is addressed as a whole elsewhere). We've also added recent (from June 2022) paper you have suggested and added information regarding succesful detection of unhydrolyzed lignan hexosides using this method.